# Influence of the Nanostitch Sensor Embedment on the Fibrous Microstructure of Glass Fiber Prepreg Laminates

**DOI:** 10.3390/polym14214644

**Published:** 2022-10-31

**Authors:** Stepan V. Lomov, Sergey G. Abaimov

**Affiliations:** Center for Petroleum Science and Engineering, Skolkovo Institute of Science and Technology, 121205 Moscow, Russia

**Keywords:** sensor embedment, nanostitch, glass fiber laminates, synchrotron radiation computed tomography, microstructure

## Abstract

Changes in the fibrous microstructure in glass fiber/epoxy prepreg quasi-isotropic laminates after the introduction of embedded sensors in the form of “nanostitch” as interleaves are investigated using 3D imaging with synchrotron radiation computer tomography (SRCT). Nanostitch interfaces are created by aligned carbon nanotubes (CNTs) with two different morphologies. The laminates are fabricated using an autoclave. The investigated microstructural features include: thickness variability of the plies and laminate, resin rich gaps at the interfaces, presence of voids, and misorientation of plies and misalignment of fibers deep inside the plies and close to the ply interfaces. The analysis of the SRCT images, at a resolution of 0.65 µm, shows the following: (1) the laminate preserves its thickness, with a resin/CNT-rich gap of ~5 µm created at the interface and the plies compacted by nano-capillarity; (2) there are no voids with sizes over 1–2 µm both in the baseline and nanostitched laminates; (3) the misorientation of plies (the in-plane difference of the average fiber direction from the nominal ply angle) is under 2°; (4) the misalignment (standard deviation of fiber orientations) has the same characteristics in the baseline and nanostitched laminates: it is in the range of 1.5°–3° in-plane and 2°–4° out-of-plane; the misalignment close to interfaces is increased in comparison with the misalignment deep within plies by ~1°. We conclude that the embedment of the nanostitch sensor does not alter the microstructural parameters of the laminate.

## 1. Introduction

Sensor embedment in composite structures with fibrous reinforcement presents a well-known problem as an increase in thickness, change in void content, and microstructure misalignment are often considered in industry as a “no go” criterion. Therefore, these issues must be carefully addressed. In this regard, a significant step forward for polymer composites manufactured by autoclave (prepreg) technology has been achieved by the implementation of nanostructures possessing a high nano-capillarity.

Vertically aligned carbon nanotubes (VACNTs, or “nanostitch”) embedded into a composite laminate interface present a known multi-functional solution for damage sensing and out-of-autoclave conductive curing [1,2,3,4] accompanied by the improvement of interlaminar fracture toughness and in-plane strength [5,6,7,8,9,10,11,12,13,14]. Mostly applied to carbon fiber laminates, VACNT interleaves were recently introduced in glass fiber laminates [15,16], which widens the range of applications to such industries as wind energy and shipbuilding.

Embedded sensors can potentially affect the micro-geometry of a composite and, hence, adversely affect its mechanical properties (see, for example, [17]). The introduction of CNTs, especially long CNTs, change the microstructure of the composite because of the high resistance of the aligned CNTs to compaction [18]. On the other hand, we argue that the oriented high nano-capillarity of the non-impregnated aligned CNT interleave may provide additional compaction pressure at the interface with the prepreg. In our study, we verify this hypothesis experimentally.

Several questions can be raised in this context:
―After the introduction of the VACNT interleave, will the thickness of laminate plies or laminate be affected?―Will the void content or void distribution change?―Will the orientations of fibers change inside plies? Closer to the interfaces?

These factors (laminate thickness, void content, and fiber (mis)orientation) have a direct relation to the laminate mechanical properties in-service:
―A possible laminate thickness change due to the embedment of interleaves has a direct relation to the interlaminar fracture toughness, as measured in the short beam shear (SBS) test [15].―Void content and void distribution strongly affect the laminate damage behavior [19].―Fiber misorientation distribution inside the plies affects the laminate mechanical properties [20,21]; fiber misorientation at the interfaces affects the interlaminar fracture toughness [22].

Micro-geometry of the nanostitched laminates was studied with a focus on the geometry of the VACNT interface. These observations have been reported in [12,13] for carbon fiber laminates and in [15] for glass fiber laminates.

The present study answers (in negative) the formulated questions for the prepreg-based, autoclave-produced quasi-isotropic glass fiber/epoxy laminates with the nanostitch sensors embedded as interleaves at all ply interfaces. Hence, the findings presented here confirm that this type of sensor can be applied in composite laminates without raising concerns about the deterioration of mechanical properties and thickness changes.

Two types of nanostitches are considered: “nanostitch 1.0” as studied in [8,10], and buckled “nanostitch 2.0” [12,13]. The microstructure of the laminates is imaged with synchrotron radiation computed tomography (SRCT) [23] and analyzed using Fiji (ImageJ) and VoxTex [24] software.

## 2. Materials

The present paper analyses the images of the laminates reported in [23]. Materials and imaging are briefly described in this section for the convenience of the reader.

### 2.1. Nanostitch

Two types of nanostitch were used: (1) “nanostitch 1.0” as VACNTs fabricated by the procedure established in [8,10], and (2) “nanostitch 2.0” as buckled arrays of densely synthesized VACNTs [12]. These two types are referred to below as “N1” and “N2”, or collectively as “Nx”. The main difference between N1 and N2 is in the geometry of CNTs: in nanostitch N1, the CNTs are vertically aligned with the nano-capillarity channels are perpendicular to ply interfaces, while in nanostitch N2, the CNTs are buckled, forming denser CNTs networks with the changes in the nano-capillarity channel directions. The reader is referred to [12] for detailed descriptions.

In both cases, catalytic chemical vapor deposition (CVD) was used for production. VACNTs were grown on silicon wafer substrates of dimensions 30 × 40 mm in a quartz tube furnace with an inner diameter of 44 mm at atmospheric pressure, as described in [8]. The wafer had uniform layers of deposited catalyst: 1 nm of Fe on top of 10 nm of Al_2_O_3_. N2 was created on wafers with patterned deposition of the catalyst, resulting in growth on an array of squares of 49 μm × 49 μm with 14 μm gaps between them. N2 forests were densified by buckling, which involved placing a bare silicon wafer on top of the silicon wafer substrate with patterned CNTs, and applying uniform pressure by hand. The height of the N1 forest was 20 µm. The N2 forest was grown to a height of 40 µm, then buckled down to a lower height.

### 2.2. Laminates and Test Samples

Prepregs Hexply^®^ 913/30%/UD192/EC9 673517 (Hexcel Composites, Daigneu, France) were used for the fabrication of laminates. E-glass fibers had diameters of 4.48 ± 0.40 µm (average and standard deviation in 50 measurements), as measured on the optical images [15] of the laminate cross sections. Nominal ply parameters, provided by the manufacturer, were as follows: resin weight fraction 30%, areal density of reinforcement 192 g/m^2^, areal density of prepreg 279 g/m^2^, density of E-glass 2.56 g/cm^3^, and density of epoxy 1.23 g/cm^3^. These data give an estimation of the nominal ply thickness of 0.129 mm.

The quasi-isotropic laminates of eight plies had a layup [90/0/45/−45]_s_. VACNT interleaves (N1 or N2) were manually transferred in-between the prepreg plies during the laminate lay-up procedure at room temperature. Randomly-oriented commercial CNT film CNTM4 (Tortech, Maalot, Israel) of 16 μm was placed on the top and bottom of the layup. The purpose of this film was to create electrical and thermal functionality of the laminate. The reader is referred to [15,16,23] for detailed information on the sample production.

Three variants of laminates were produced: baseline (BSL) without interleaves; N1 with N1 interleaves at all ply interfaces; and N2 with N2 interleaves at all ply interfaces (naming laminates by the names of the embedded interleaves leads to no confusion in the following text).

The laminates of all three types were cured in an autoclave, following the manufacturer-recommended cure cycle, as follows:
―Increase pressure from vacuum at 0.07 MPa to 0.6–0.7 MPa and hold throughout the cure cycle. Vent the vacuum when the pressure inside chamber reaches 0.14 MPa.―Increase temperature at a steady rate of not more than 5 °C/min until reaching 90 °C.―Dwell: Hold the temperature at 90 °C (195 F) for 30 min.―Increase the temperature at a steady rate of not more than 5 °C/min until 120–130 °C and hold for 60 min.―Decrease the temperature at a rate below 3 °C/min; release the pressure to 0 MPa when the temperature reaches 65 °C.

The nominal expected thickness of the laminates was 0.129 × 8 = 1.032 mm. Plates of 150 × 150 mm were fabricated in the autoclave: a BSL laminate plate and plates with regions of 30 × 50 mm for the nanostitched laminates. The nanostitched regions were cut into Double End Notch Test (DENT) specimens, which were imaged during in situ loading inside the synchrotron test chamber. The present paper discusses only the results of the pre-load imaging. The in situ DENT results have been published elsewhere [23].

### 2.3. Synchrotron Radiation Computed Tomography

SRCT imaging was performed at the European Synchrotron Radiation Facility, beamline ID19. The following parameters were used: 20 keV energy X-ray beam, 50 ms exposure time, 60 mm propagation distance for enhanced phase contrast at interfaces, 2996 radiographic projections over 180° range, 0.65 μm isotropic voxel size, and 1.6 × 1.6 × 1.4 mm^3^ field-of-view. The imaging was done on the DENT specimens; for each variant of laminates (BSL, N1, or N2), the imaging was done on two specimens, at two locations on each specimen.

The fibrous ply orientation orthogonal to the axis of rotation of a sample in the micro-tomography may lead to the appearance of artefacts in the image of the ply [25]. The artefacts can be diminished if the so-called “canted” positioning of the sample in the loading rig is used [26]. Figure 1 shows the arrangement of the imaging. The DENT sample (Figure 1a) is installed into the test rig (Figure 1b), which is canted—the test rig, and, hence, the sample rotation axis, are inclined by 15° to the vertical, and thus no plies in the laminate are parallel to the beam direction anymore. Figure 1c shows schematically the central region of the DENT specimen and the positions of two images, P1 and P2 (overlapped for the following merge in software).

The coordinate system XYZ of the images and its difference from the laminate coordinate system (denoted as 0°90°) is illustrated in Figure 1c. Axis X represents the out-of-plane direction of the laminate, it is orthogonal to the middle surfaces of plies and their interfaces, which are parallel to the YZ coordinate plane. The Z and Y directions correspond to the 0° and 90° laminate directions, respectively, rotated by 15° counter-clockwise around the X axis.

For each laminate variant, two samples were scanned, and for each sample, two 3D images (P1 and P2, Figure 1c) were taken. After reconstruction, the 3D image was a stack of 2159 TIFF slices, uncompressed, 32 bit, dimensions of 2560 × 2560 × (2159) pix^3^, and the corresponding file size was ~25 MB. The slices were parallel to XY coordinate plane, and they were stacked in the Z direction.

Before further analysis, the images were processed using the functionality of VoxTex as follows:
Cropping: The images were cropped to dimensions ~1550 × 1400 × (2159) pix^3^ so that they did not contain air surrounding the sample. The exact XY dimensions varied from image to image. The resulting files were ~9 MB size.Density scaling: The actual grey scale range of the images is shorter than the full range from black to white. Therefore, the range was rescaled so that black (0) corresponds to the lowest grey scale value, and white (255) corresponds to the highest grey scale value of the image.Filtering: The Gaussian filter with kernel size 3 and standard deviation 2.0 pix was applied to the images. This operation eliminated high frequency noise and is recommended before analyzing the image anisotropy and directionality.

Figure 2 shows typical images for three variants of laminates. Ply angles, representing nominal fiber directions in the laminate coordinate system, are shown. Note the difference with the image coordinate system, rotated by 15° around the X axis (horizontal on Figure 2). Note also the difference in the appearance of +45° and −45° angle plies in the BSL image, on the one hand, and in the N1 and N2 images, on the other. This was caused by mirroring the sample when installing it on the rig: accidentally, the BSL specimen was canted in the direction opposite to the directions in which the Nx samples were canted, hence the tilt went in a different direction with respect to the 45° angle. This difference was accounted for in the processing of the fiber directions. The nominal ply angles will be used systematically in the paper for the identification of plies.

The film on the surface of the laminate, shown in Figure 2 for N2 as the inset, makes the identification of the boundary of the outer plies uncertain, and the thickness of these plies is not well defined. Therefore, the film on the surface of the laminate is not analyzed in the present study and the thickness of the outer of 90° plies is also excluded from the data processing.

For each laminate variant, four 3D images (two samples and two positions in each sample) were analyzed. They were designated as BSL*ij* or Nx*ij*, where BSL or Nx means laminate variant and *ij* means sample *i*, position *j*, with *i,j* taking values 1 or 2. For example, N111 is N1 laminate, sample 1, and position 1.

## 3. Fibrous Microstructure of Laminates

The following parameters of the microstructure were investigated:
―thickness of plies and of the inter-ply interfaces (and, thereby, of the laminate);―presence of voids;―(mis) orientation of fibers deep inside the plies and closer to their interfaces.

In the subsections of this section, the measurement methodology of these parameters is presented first, followed by the results of the characterization.

### 3.1. Thickness of Plies and Interfaces

#### 3.1.1. Measurement Method

Figure 3 illustrates the method for measuring the plies thicknesses. The thickness of a ply was measured as the distance between two parallel lines, fitted manually along the left and right boundaries of the ply (as shown in Figure 3). Positions of the boundaries were chosen based on closeness to the fibers of the ply. The boundary fitting shown in Figure 3 was performed on five slices, positioned with even spacing along the Z axis (images 400, 800, 1200, 1600, and 2000). These five measurements were averaged for the considered ply; the differences among them were within ±5 µm.

For BSL (no interleaves), interfacial gaps are absent, and the boundary of a ply serves as the boundary of the adjacent ply. For Nx, interfacial gaps, filled with VACNTs, exist between the plies, which were identified by two parallel boundaries, separated by the gap distance. If a boundary or gap between two central −45° plies was not visible, as shown in Figure 3 (in BSL due to fiber nesting and sometimes even in N1 and N2 due to the fact that interleave not always prevents nesting in-between two identically oriented plies), the boundaries between the plies were not identified and the plies’ thicknesses were calculated as half of that measured for the two plies together. The outer 90° plies did not have well-defined boundaries; therefore, they (and the interleaves adjacent to them) were excluded from the ply and laminate thickness analysis.

#### 3.1.2. Results

Table 1 shows results of the thickness measurements. Several observations may be stated:
―*Variability:* The individual ply thicknesses vary, as can be observed in Figure 3. These variations (standard deviation) are ~6% for BSL, ~10% for N1, and ~8% for N2. The nanostitch seems to increase ply thickness variability, but the difference in variance between BSL and Nx corresponds to statistical significance of only ~0.8. Ply thickness may be expected to be affected, in principle, by the ply position in a laminate. This was not, however, the case.―*Nominal ply thickness:* The averaged value of 121 µm for the BSL ply thickness differs from the nominal value of 129 µm.―*Gap:* The average interface gap in Nx laminates (as interleave thickness) equals 5.2 µm for N1 and 3.3 µm for N2, i.e., is comparable to the fiber diameter.―*Difference between BSL and Nx:* The mean ply thickness of the Nx laminates is lower than for BSL by 3–5 µm, which is approximately the interleave thickness in the Nx laminates. In the result, the embedment of the nanostitch interleaves at every interface between plies does not cause a change in the laminate thickness on average. It seems that the nanostitch due to its nano-capillarity provides additional pressure that compresses fibrous plies, maintaining an overall unchanged laminate thickness (more accurately, the difference in means does not pass a statistical test for significance because of a high variance). We observe no significant difference in this effect between N1 and N2 laminates, suggesting that the direction of the nano-capillarity channels does not play a significant role.―*Fiber volume fraction inside the plies:* VF = 62.0% for BSL and 64.6% for Nx, serving again as an indication of the additional ply compression introduced by the nanostitch.

### 3.2. Voids

There were no voids visible when the images were visually inspected. However, certain artefacts, which could be taken for micro-void presence, were detected on the images. These artefacts are illustrated in Figure 4. They were seen on the images as black spots of 5–10 µm in size, located next to the fibers. When the image was segmented with the threshold grey value of 30 (black is 0 and white is 255; the threshold value was determined based on the analysis of the grey scale histograms), the mentioned artefacts were features with a size significantly below 1000 µm^3^ and an average aspect ratio below 10. Such features could be attributed to micro-voids. However, several reasons have led to conclusion that these features were imaging artefacts. As it is seen in Figure 4b,c, the “voids” were seen/not seen in specific plies with the same actual fiber orientation in BSL and N1, even if these plies were positioned differently in the laminate thickness (cf. Figure 2): in the inner plies oriented at 60° to the rotation axis, there were no “voids” seen, and low “voids” numbers were observed in the outer plies, oriented at 75° to the rotation axis. The existence of real voids would not depend on the ply orientations. Moreover, the “voids” were observed only from one “lower” side of the fibers (see Figure 4a). Finally, the “voids” were not seen in optical micrographs with a higher resolution, see [15]. Such artefacts on the boundaries of “white” regions were also seen in images, as shown in [27]; the scan orientation artefacts in relation to voids are discussed in [28].

The conclusion is that there were no voids observed in both the BSL and Nx laminates.

### 3.3. Fiber Orientation

#### 3.3.1. Measurement Method

The identification of fiber orientation was done in *VoxTex* software (KU Leuven, Belgium) based on the structure tensor analysis of the 3D images. The method was described in detail in [24]. Structure tensor calculations were validated in comparison with synthetic images in [24] and with high-fidelity analysis of micro-CT images using AVIZO software [29,30].

The procedure described in [25] was used for the analysis. The software transforms an SRCT image into a *voxel* model, with voxels being cubic volumes 10 × 10 × 10 pixels. For each voxel, using a structure tensor method with an integration window radius of 8 pixels, the following parameters were determined:
―degree of anisotropy *β* ∈ [0, 1], where *β* = 0 corresponds to the isotropic image structure (pictured by black color) and *β* = 1 corresponds to the transversely isotropic material (pictured by white color);―principal direction of anisotropy *p*, which for voxels corresponding to the fiber reinforced regions was identified with the fiber direction;―average grey scale of the voxel.

Degree of anisotropy was used to determine the voxels corresponding to the fiber reinforcement. Only these voxels were used for the definition of the fiber orientation.

Figure 5 illustrates selection of the anisotropy threshold. A region of interest (ROI) was selected fully inside a ply (Figure 5a). The distribution of *β* within the ROI (Figure 5b) and its histogram (Figure 5c) showed high values of *β* in ROI, with occasional voxels, corresponding to inter-fiber matrix micro-pockets. The data, presented in Figure 5, were typical for plies of different orientations in the BSL and Nx laminates. Based on these data, a value of *β** = 0.8 was chosen as a threshold for the orientation calculations.

The fiber orientation vector *p* defines two fiber orientation angles, which are spherical angles in the laminate coordinate system, denoted as 0°90° in Figure 1. The laminate coordinate system is the image system *YZX*, rotated by 15° around the X axis. Spherical angle *φ* represents the in-plane and angle θ represents the out-of-plane fiber orientations.

For a ply with a nominal fiber orientation *φ_nom_* (as equaling either 0°, or ±45°, or 90°), the following parameters describe the variability of the fiber orientation:
―ply in-plane misorientation <*φ*> −*φ_nom_*, where <…> designates the averaging of the fiber orientations over the ROI;―ply out-of-plane misorientation <*θ*> −90°;―fiber in-plane misalignment Δ*φ* = *std* (*φ*), where *std* () is standard deviation over the ROI;―fiber out-of-plane misalignment Δ*θ* = *std*(*θ*), where *std* () is standard deviation over the ROI.

Two types of ROI were used for the characterization of the variability of the fiber directions: “ply ROI” as fully placed inside the ply and “interface ROI” as placed over ply’s interface at both sides of the ply and having width of 60 pix = 39 µm (hence, covering about three fibers in the thickness direction from the interface, both sides). In the interface ROI, *φ*-related parameters were defined separately for two opposite interfaces of the ply.

#### 3.3.2. Results

Figure 6 illustrates a typical result of the orientation analysis, for in-plane and out-of-plane, for the example of ply 0° (Figure 6b–d) and interface 90°/0° (Figure 6a–f) in a BSL image. The mean and standard deviation (RMS), indicated on the graphs, characterize the ply misorientation and fiber misalignment, respectively. In the case of in-plane orientation on the interface (Figure 6e), two histogram peaks appear; in this case, the mean and standard deviation are calculated separately for two parts of the distribution and the average of two is used for the final characterization.

In-plane and out-of-plane misorientations of fibers for all images lie within the range of ±3°. Figure 7 shows a summary for all types of laminates. There is no trend in <*φ*> −*φ_nom_* (Figure 7a) and no systematic difference between the BSL and Nx data. This means that the layup was done with a good orientation quality and in a stable way for all of the samples. A weak downward trend in <*θ*> −90° values (Figure 7b) across the laminate thickness indicates a slight inclination of the plies. This is observed for all of the samples.

Figure 8 shows fiber misalignment angles in the plies and interfaces. Several observations can be made, as follows:
―*Data quality:* The data are consistent and repeatable in four images for all three laminate variants.―*Misalignment deep inside plies:* Average in-plane/out-of-plane misalignment in plies does not exceed 3°/4°, respectively; when averaged over all of the plies, it is on the level of 2°/3°, respectively.―*Misalignment at interfaces:* Normally higher than inside the neighboring plies by about 1° both for the in- and out-of-plane angles. The fiber misalignment is illustrated in Figure 9, which shows in-plane slices in 0° and 45° plies and at their interface. In-plane misalignment at interfaces with 45° difference of plies orientations (interfaces 0/45°) is about 1° larger than at the interfaces of plies with orthogonal orientations (interfaces 0°/90° and +45°/−45°). For the out-of-plane misalignment at the interfaces, this difference is not observed.―*Difference between the baseline and nanostitched laminates:* The features, stated above, are seen in both the BSL and Nx laminates. No systematic difference is observed between the misalignment angles in three laminates, neither for in-plane nor for out-of-plane angles.

It can be concluded that the nanostitch does not influence the ply misorientation and fiber misalignment.

## 4. Conclusions

The introduction of the nanostitch does not alter the microstructural parameters of the glass fiber-reinforced epoxy quasi-isotropic laminate, produced in an autoclave.

The following features of the baseline and nanostitched laminates have been observed in the SRCT images:
After the nanostitch embedment, the laminate preserves its thickness, with resin/CNT-rich gaps of ~5 µm created at plies’ interfaces and corresponding to a decrease in individual ply thicknesses. We argue that the latter effect is related to the effective pressure created by the nanostitch nano-capillarity.There are no voids with sizes over 1–2 µm observed, both in the baseline and in the nanostitched laminates.The misorientation of plies (as the difference of the average fiber direction in a ply and the nominal ply angle) is under 2°.The in-plane and out-of-plane misalignment (standard deviation of fiber orientation) inside the plies is in the range of 1.5°–3° and 2°–4°, respectively; the misalignment closer to the plies interfaces is increased in comparison with the misalignment within plies by ~1°.

The observed stability of the microstructural parameters of the laminate after the embedment of the nanostitch ensures that the laminate with the embedded sensor fully preserves its geometrical and mechanical properties. The measurement of the short beam strength of the nanostitched laminates, reported elsewhere [15,16,23], fully confirms this conclusion.

## Figures and Tables

**Figure 1 polymers-14-04644-f001:**
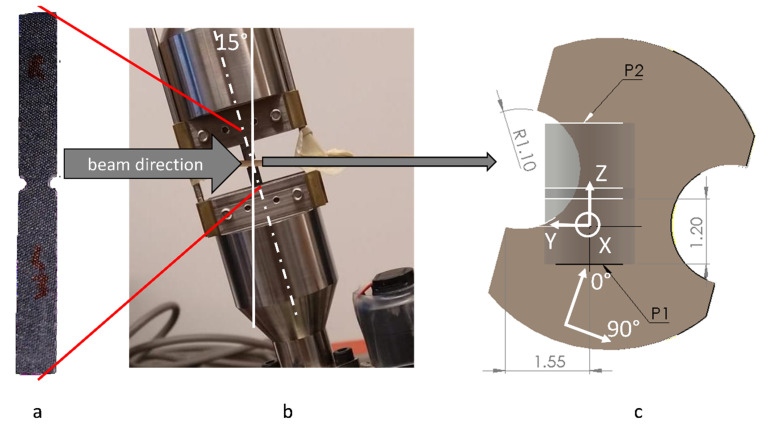
A canted SRCT rig: (**a**) DENT specimen; (**b**) specimen arrangement on a loading stage; (**c**) schematics of the image locations (P1 and P2, overlapped).

**Figure 2 polymers-14-04644-f002:**
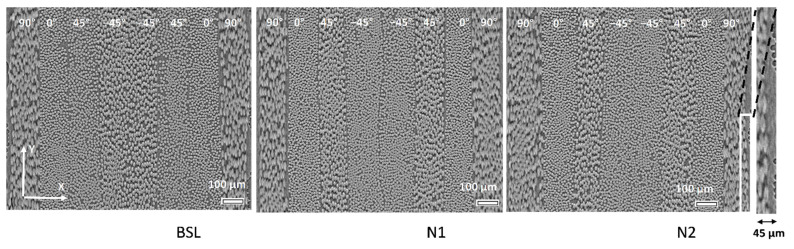
Images of the laminates: XY slices, slice # 1000 for all of the images. Nominal ply angles are shown. Right: close-up on the film region on the surface of the laminate.

**Figure 3 polymers-14-04644-f003:**
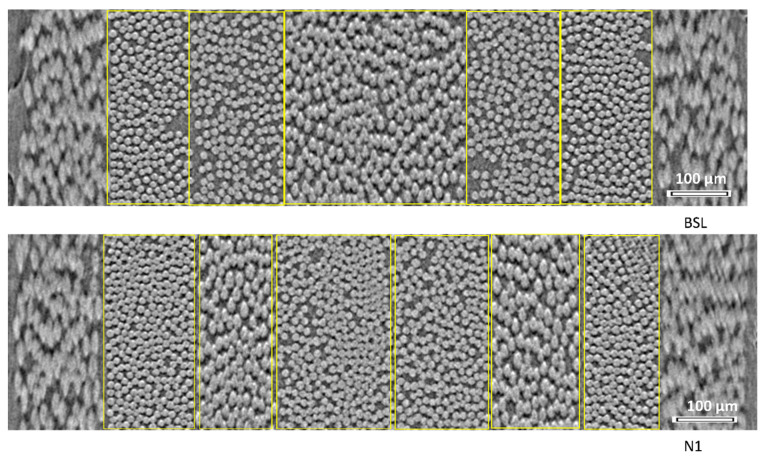
The scheme of measurement of plies’ thicknesses (measurement for N2 is similar to N1 and not illustrated).

**Figure 4 polymers-14-04644-f004:**
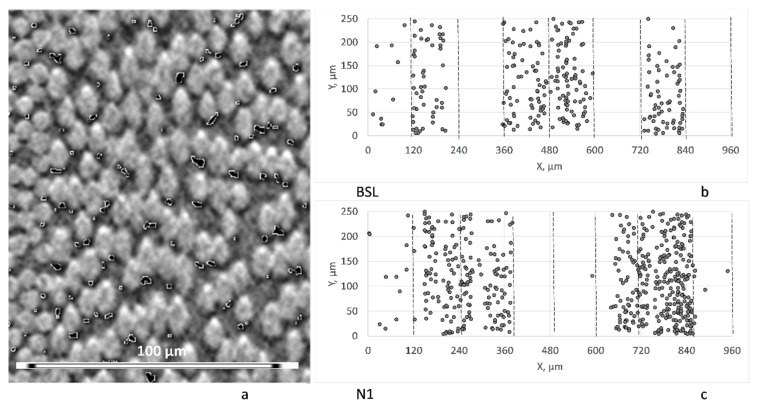
Voids-like artefacts: (**a**) artefacts indicated by boundaries of a 30-threshold level; (**b**,**c**) positions of artefacts in the BSL (**b**) and N1 (**c**) images; dashed lines indicate plies’ boundaries.

**Figure 5 polymers-14-04644-f005:**
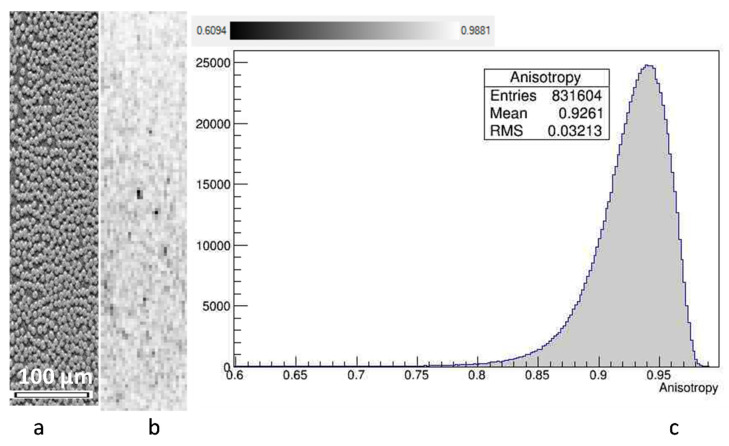
Anisotropy inside plies, a typical case (BSL11, ply −45°): (**a**) a slice of the ROI; (**b**) distribution of degree of anisotropy in voxels (with isotropy presented as black color and anisotropy as white color); (**c**) histogram of the degree of anisotropy.

**Figure 6 polymers-14-04644-f006:**
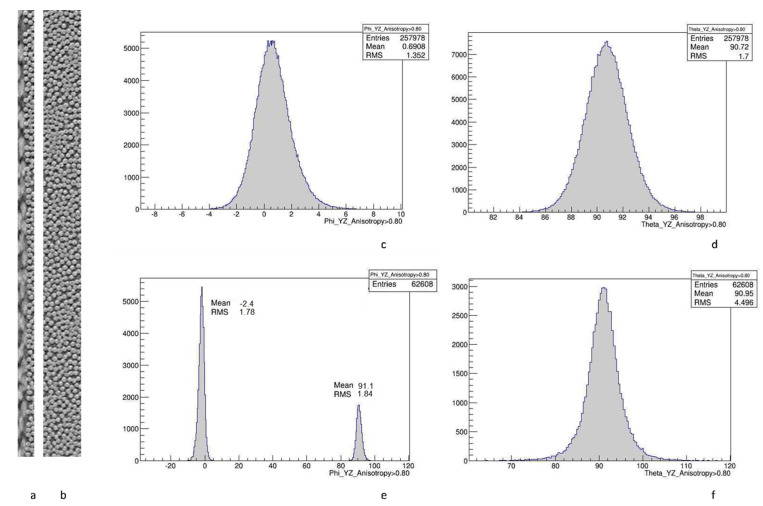
Typical fiber orientation analysis, image BSL21: (**a**) interface 90°/0°; (**b**) inside ply 0°; (**c**) in-plane fiber orientation angle, inside 0°; (**d**) out-of-plane fiber orientation angle, inside 0°; (**e**) in-plane fiber orientation angle, interface 90°/0°; (**f**) out-of-plane fiber orientation angle, interface 90°/0°.

**Figure 7 polymers-14-04644-f007:**
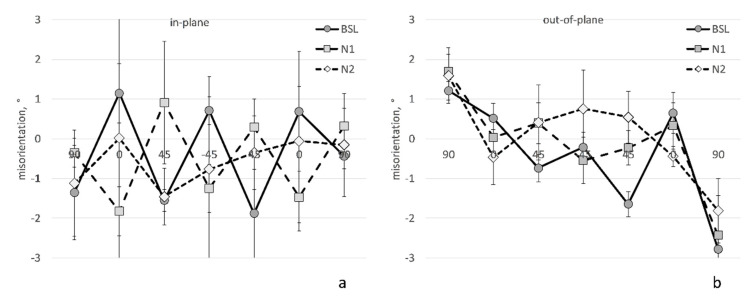
Misorientation of plies: (**a**) in-plane; (**b**) out-of-plane. Horizontal axis enumerates values of the nominal ply orientations. Error bars are standard deviations in four 3D images.

**Figure 8 polymers-14-04644-f008:**
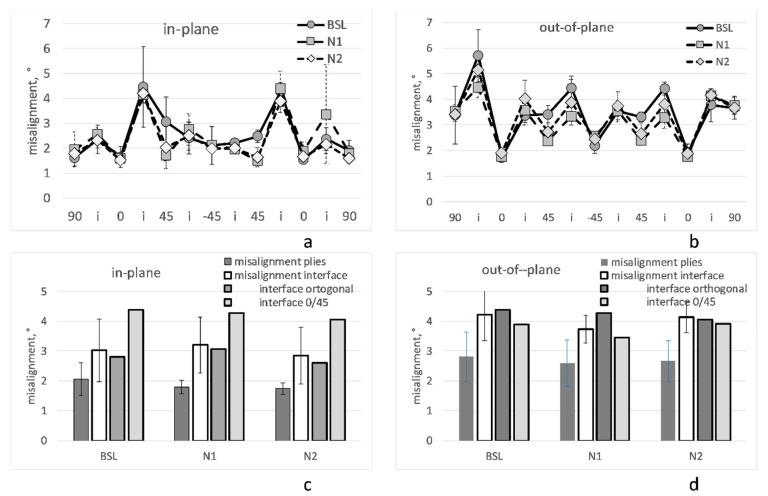
Misalignment of fibers deep inside plies and at the interfaces: (**a**,**b**) misalignment of in-plane (**a**) and out-of-plane (**b**) angles; horizontal axis enumerates values of the nominal ply orientations and interfaces (“i”) in-between the plies; (**c**,**d**) average values of in-plane (**c**) and out-of-plane (**d**) misalignment angles inside plies and at the interfaces.

**Figure 9 polymers-14-04644-f009:**
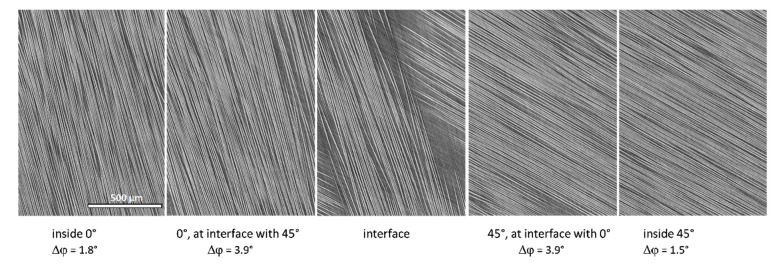
In-plane fiber misalignment over the 0°/45° interface, laminate N111. The images are rotated 15° counter-clockwise. The scale is the same for all of the images.

**Table 1 polymers-14-04644-t001:** Thickness of plies and interfacial gaps, mean, and standard deviation, µm.

	Plies 0°	Plies 45°	Plies −45°	All Plies	Gap
BSL	120 ± 10	121 ± 4	122 ± 4	121 ± 7	n/a
N1	117 ± 12	112 ± 9	119 ± 12	116 ± 11	5.2 ± 0.6
N2	114 ± 9	120 ± 6	118 ± 11	117 ± 9	3.3 ± 0.9

## Data Availability

Restrictions apply to the availability of the data used in this article.

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
