# Peer review of "Influence of the Nanostitch Sensor Embedment on the Fibrous Microstructure of Glass Fiber Prepreg Laminates"

_polymers, 2022, doi:10.3390/polym14214644_

Round 1

Reviewer 1 Report

In this paper, the authors report the effect of the introduction of vertically aligned carbon nanotubes (nanostitch) in polymer composite laminates. This manuscript demonstrates interesting results, but needs some modifications to be published in "Polymers". My comments are as follows:

1. It is not appropriate to express the title in a sentence and it needs to be corrected.

2. Keywords should be selected among the words used in the Abstract.

3. In Figures 7 and 8(a,b), please express the symbol for each sample as different. In particular, the distinction between BSL and N1 is poor.

Author Response

polymers-1953460          Lomov S.V., Abaimov S.G

ANSWERS TO REVIEWERS

Reviewer 1

Reviewer’s comment

Author’s answer

In this paper, the authors report the effect of the introduction of vertically aligned carbon nanotubes (nanostitch) in polymer composite laminates. This manuscript demonstrates interesting results, but needs some modifications to be published in "Polymers".

We appreciate a positive evaluation of our results, and have done the proposed modifications in the revised manuscript.

1. It is not appropriate to express the title in a sentence and it needs to be corrected.

The title is changed grammatically, as suggested by the reviewer, and also to emphasize sensor embedment application

2. Keywords should be selected among the words used in the Abstract.

The keywords changed as suggested by the reviewer

3. In Figures 7 and 8(a,b), please express the symbol for each sample as different. In particular, the distinction between BSL and N1 is poor.

The figures are redrawn as suggested by the reviewer.

Reviewer 2 Report

Lomov et al. studied whether nano-stitched interfaces affect the fibrous microstructure of glass fiber prepreg laminates. Two types of nano-stitch interfaces are considered. The microstructure of the laminates is imaged with synchrotron radiation computed tomography (SRCT), and analyzed using Fiji (ImageJ) and VoxTex software. The key finding is that introducing the nano-stitch does not alter the microstructural parameters of the laminate. Overall, this manuscript is publishable but must address the following concerns:

1.       The author should improve their English. I read it several times before I understood the author's intention.

2.       What is the meaning of this study? Why did the author study the effect of nano-stitch on the fibrous microstructure? I only see the author who presents the results. I suggest the author can better clarify this work's meaning.

Author Response

polymers-1953460          Lomov S.V., Abaimov S.G

ANSWERS TO REVIEWERS

Reviewer 2

Reviewer’s comment

Author’s answer

Lomov et al. studied whether nano-stitched interfaces affect the fibrous microstructure of glass fiber prepreg laminates. Two types of nano-stitch interfaces are considered. The microstructure of the laminates is imaged with synchrotron radiation computed tomography (SRCT), and analyzed using Fiji (ImageJ) and VoxTex software. The key finding is that introducing the nano-stitch does not alter the microstructural parameters of the laminate. Overall, this manuscript is publishable but must address the following concerns:

We thank the reviewer for the clear summarising the essence of our work. We have done the proposed modifications in the revised manuscript.

The author should improve their English. I read it several times before I understood the author's intention.

The text was re-written extensively and corrections in English grammar were made.

What is the meaning of this study? Why did the author study the effect of nano-stitch on the fibrous microstructure? I only see the author who presents the results. I suggest the author can better clarify this work's meaning.

In the revision, additions are made in Introduction and Conclusions sections, which explain in more details the aims and the context for the study - the influence of the effect of the sensor (nanostitch) embedment on the laminate microstructure.

Reviewer 3 Report

1, the language expression in the manuscript should be improved;

2, the title of manuscript can't reflect the  innovation of this work;

3, what are the characteristical differences “N1” and “N2”;

4, the experimental results should be described detail;

5, what are the reasons for the conlusion should decribed theorically.

Author Response

polymers-1953460          Lomov S.V., Abaimov S.G

ANSWERS TO REVIEWERS

Reviewer 3

Reviewer’s comment

Author’s answer

1, the language expression in the manuscript should be improved;

The text was extensively re-written and corrections in English grammar were made.

2, the title of manuscript can't reflect the  innovation of this work;

The title is improved to emphasize the sensor (nanostitch) embedment as the problem addressed

3, what are the characteristical differences “N1” and “N2”;

The explanatory paragraph is added in section 2.1

4, the experimental results should be described detail;

Experimental results on the image processing and geometry change due to nanostitch embedment are discussed in detail. Experimental results on the change in mechanical properties are discussed elsewhere, and references to the detailed description were added.

5, what are the reasons for the conclusion should described theatrically.

We have changed the conclusions addressing that no changes happen due to sensor embedment.

Round 2

Reviewer 2 Report

I suggest publication after reducing the self-citation rate. Currently it is  44%, which is too high. 

Author Response

The self-citation was significantly reduced by replacing it with other references.

The extensive English polishing was conducted.

Reviewer 3 Report

this manuscript is suitable for publication.

Author Response

English was polished extensively in the manuscript.